# Evaluation of Anthocyanin Profiling, Total Phenolic and Flavonoid Content, and Antioxidant Activity of Korean *Rubus* Accessions for Functional Food Applications and Breeding

**DOI:** 10.3390/antiox14081012

**Published:** 2025-08-18

**Authors:** Juyoung Kim, Jaihyunk Ryu, Seung Hyeon Lee, Jae Hoon Kim, Dong-Gun Kim, Tae Hyun Ha, Sang Hoon Kim

**Affiliations:** 1Advanced Radiation Technology Institute, Korea Atomic Energy Research Institute, 29 Geumgu-gil, Jeongeup-si 56212, Jeollabuk-do, Republic of Korea; jykim83@kaeri.re.kr (J.K.); jhryu@kaeri.re.kr (J.R.); lshyeon@kaeri.re.kr (S.H.L.); jaehun@kaeri.re.kr (J.H.K.); dgkim-@kakao.com (D.-G.K.); hath1002@kaeri.re.kr (T.H.H.); 2Jangheung Research Institute for Mushroom Industry, Jangheung 59338, Jeollanam-do, Republic of Korea; 3Department of Horticulture, Kongju National University, 54 Daehak-ro, Yesan-gun 32439, Chungcheongnam-do, Republic of Korea

**Keywords:** *Rubus*, berries, anthocyanin, antioxidant activity, fruit coloration

## Abstract

The *Rubus* genus includes numerous berry species known for their rich phytochemical content and antioxidant properties. However, comparative evaluations of wild and cultivated *Rubus* germplasms in East Asia remain limited. This study aimed to identify superior resources with potential for use in functional foods and breeding through integrated phytochemical and antioxidant profiling. Fifteen accessions collected across Korea were assessed for fruit coloration, total phenolic content (TPC), total flavonoid content (TFC), five antioxidant activities (DPPH, ABTS^+^, superoxide, ferric-reducing activity power, and Fe^2+^ chelation), and anthocyanin composition by high-performance liquid chromatography‒Mass spectrometry. The TPC ranged from 1.03 to 7.54 mg g^−1^ of frozen fruit, and TFC ranged from 2.75 to 7.52 mg g^−1^ of frozen fruit, with significant differences among accessions (*p* < 0.05). Black-colored fruits such as *R. coreanus* and *R. ursinus* varieties exhibited high anthocyanin levels (approximately total 471 and 316 mg g^−1^ extracts, respectively), with cyanidin-O-hexoside and cyanidin-3-O-glucoside being the dominant pigments. However, the antioxidant performance of these accessions varied. A wild *R. crataegifolius* (no. 9, resource F) showed the highest TPC and ranked within the top five in multiple antioxidant assays, despite its moderate anthocyanin content. Correlation analysis revealed that TPC and TFC were significantly associated with antioxidant activity (*p* < 0.05) but not directly with anthocyanin content. These results suggest that antioxidant potential is influenced by a broader spectrum of phenolic compounds, rather than anthocyanins alone. These findings underscore the need to look beyond visual traits and focus on biochemical evidence when selecting elite *Rubus* accessions.

## 1. Introduction

Wild raspberry is a fruit belonging to the *Rubus* genus, typically harvested from deciduous shrubs growing in the wild. The *Rubus* genus encompasses over 700 species worldwide, such as commonly cultivated raspberries (*Rubus idaeus* L. and subspecies) and blackberries (*Rubus occidentalis* L. and subspecies) [1]. These berries are valued not only for their taste and aroma but also for their abundance of functional phytochemicals such as anthocyanins and ellagic acids as well as essential nutrients like carbohydrates and amino acids [2,3]. Among these, phenolic compounds in *Rubus* fruits are particularly recognized for their potent antioxidant activities. These bioactive constituents are attractive to consumers seeking dietary, nutritional, and therapeutic benefits, as *Rubus* berries have long been used in pharmacological contexts [4,5].

Berry fruits, including members of *Rubus*, are rich in various phenolic compounds such as flavonoids (e.g., anthocyanins), phenolic acids, and tannins (e.g., ellagitannins), all of which are known to contribute significantly to antioxidant properties [3,6,7]. In particular, anthocyanins and ellagitannins are considered as major contributors to the antioxidant capacity of *Rubus* fruits [6]. These compounds not only affect fruit coloration but also serve as important biomarkers of antioxidant potential. For instance, a study on 19 raspberry cultivars found strong correlations between radical scavenging activity and both total phenolic and ellagic acid contents, highlighting the importance of phenolic concentration in determining antioxidant strength [8].

The bioactivities of *Rubus* fruits have been validated in numerous studies. Total phenolic and anthocyanin contents, alongside various antioxidant activity assays, serve as key indicators for evaluating their potential as functional food ingredients. Generally, *Rubus* berry resources with higher phenolic content tend to exhibit strong antioxidant activities. One study [9] analyzed six berry species, including the Korean native *Rubus coreanus*, and reported significantly higher levels of total phenolics, flavonoids, and anthocyanins in *R. coreanus* compared to other berries. This species also exhibited superior performance in antioxidant assays such as 2-2-diphenyl-1-picrylhydrazyl (DPPH), 2,2′-azino-bis(3-ethylbenzothiazoline-6-sulfonic acid) cation (ABTS^+^), and ferric-reducing activity power (FRAP) radical scavenging. Similarly, another study [8] examined 19 raspberry cultivars and found substantial variation in phenolic, anthocyanin, and ellagic acid contents, all of which were highly correlated with radical scavenging capacities. These findings emphasize the genetic diversity within the *Rubus* genus and its influence on functional compound composition and antioxidant properties.

Despite the growing interest in *Rubus* fruit research, comparative studies involving both wild and cultivated *Rubus* germplasms are still limited, particularly in East Asia. In addition, bioactive compounds responsible for antioxidant properties in *Rubus* fruits are also closely related to therapeutic or pharmacological properties. These include anti-inflammatory, antimicrobial, anti-obesity, anticancer, and antidiabetic activities, which have been validated in both in vitro and in vivo models [5,10,11]. Such recent findings have elevated *Rubus* species as promising candidates not only for functional food development but also for considerable pharmacological applications. Therefore, a deeper understanding of their phytochemical and antioxidant properties is crucial for selecting elite accessions with superior biofunctional potential.

In this study, we investigated 15 *Rubus* accessions, including both cultivated and wild types collected across Korea. We conducted chromaticity analysis of fruit coloration, quantified total phenolic and flavonoid contents, assessed antioxidant capacity using five assays (DPPH, ABTS^+^, superoxide radical scavenging, FRAP, and Fe^2+^ chelation), and profiled anthocyanin compounds using a high-performance liquid chromatography–mass spectrometer (HPLC-MS). Through this comprehensive phytochemical and bioactivity evaluation, we aimed to identify superior *Rubus* accessions with high functional potential to support the development of functional food ingredients and breeding resources. To our knowledge, this is one of the few studies that systematically compares both wild and cultivated *Rubus* accessions from Korea.

## 2. Materials and Methods

### 2.1. Collection and Preparation of Genetic Resources

The *Rubus* genetic resources listed in Table 1 were collected from wild populations or obtained via donation from the breeder Hanjik Cho. Collection sites (as DMS decimal degrees) in Korea, including Wanju (35.984182, 127.251835), Jeongeup (35.498701, 126.835837), and Namwon (35.446332, 127.535857). The berries were harvested between 20 June and 10 July 2022, and were frozen.

All measurements described below were performed in triplicate (*n* = 3) to ensure statistical reliability and minimize experimental variability. This level of replication for phytochemical and antioxidant assays in *Rubus* fruits provides sufficient statistical power to detect significant differences among accessions.

### 2.2. Color Value Measurement

A total of 10 g of frozen berries was extracted by agitation in 100 mL of pure methanol at 10 °C in the dark. The color of the extracts was evaluated using a color spectrophotometer (ColorMate; Scinco, Seoul, Korea) after filtering with Whatman grade 2 paper (GE healthcare, Chalfont St Giles, UK). Measurements were performed in triplicate, and color values (L* for lightness, a* for redness, and b* for yellowness) were analyzed using ColorMaster Software, version 3.6 (Scino, Seoul, Korea) following the Hunter’s color scale [12].

### 2.3. Analysis of Total Phenolic and Flavonoid Content

After harvesting the berries, 10 g of frozen berries were extracted by agitation with 100 mL of pure methanol in a dark room at 10 °C. After filtering with Whatman grade 2 paper (GE healthcare, Chalfont St Giles, UK), this extract was used to measure total phenolic content (TPC) and total flavonoid content (TFC) and conduct subsequent radical scavenging assays. All TPC and TFC measurements were performed in triplicate to ensure reproducibility.

TPC was assessed using the Folin‒Denis method. Briefly, 50 µL of extract was mixed with 50 µL of 1 N Folin‒Ciocalteu’s reagent (Sigma-Aldrich, St. Louis, MO, USA) in a 96-well plate, vortexed, and incubated at room temperature for 5 min. A total of 50 µL of 10% (*w*/*v*) sodium carbonate (Na_2_CO_3_; Sigma-Aldrich, St. Louis, MO, USA) solution was added, and the mixture was incubated for 1 h. Absorbance was measured at 725 nm using a microplate reader (Multiskan SkyHigh; Thermo Scientific, Waltham, MA, USA) against a methanol blank. Tannic acid (Sigma-Aldrich, St. Louis, MO, USA) was used to generate a standard curve as a standard (0 to 1 mg mL^−1^, *R*^2^ > 0.99), and the results were expressed as tannic acid equivalents (TAE, mg g^−1^).

TFC was measured based on a method established by [13]. Briefly, 20 µL of extract was mixed with 20 µL of 1% (*w*/*v*) sodium nitrite (Sigma-Aldrich, St. Louis, MO, USA) and 160 µL of 60% (*v*/*v*) ethanol in a 96-well plate and was incubated for 6 min at room temperature. Then, 15 µL of 5% (*w*/*v*) sodium hydroxide (Sigma-Aldrich, St. Louis, MO, USA) solution was added, and the mixture was allowed to react for 30 min in the dark. Absorbance was measured at 405 nm using the same microplate reader against a methanol blank. Quercetin (Sigma-Aldrich, St. Louis, MO, USA) was used to generate a standard curve as a standard (0 to 1 mg mL^−1^, *R*^2^ > 0.99). The results were expressed as quercetin equivalents (QE, mg g^−1^) using the following equation: TPC or TFC = C × V × M^−1^, where C is the standard (tannic acid or quercetin) concentration from the calibration curve, V is the extract volume, and M is the sample mass.

### 2.4. Anthocyanin Analysis Using High-Performance Liquid Chromatography–Mass Spectrometry (HPLC-MS)

Harvested berries were extracted in methanol containing 1% formic acid at 4 °C in the dark for 24 h. Extracts were filtered through a 0.45 µm membrane filter. A total of 10 µL from each of three extract replicates was analyzed using an HPLC system (Agilent 1260 series; Agilent Technologies, Santa Clara, CA, USA) equipped with an automatic injection module and quadrupole MS (Agilent 6130; Agilent Technologies, Santa Clara, CA, USA). The Poroshell 120 SB-C18 column (150 × 4.6 mm i.d., 2.7 µm particle size; Agilent Technologies, Santa Clara, CA, USA) and a compatible C18 guard column (4 × 3 mm i.d.; 3 µm particle size; Phenomenex, Torrance, CA, USA) were used for separation. Anthocyanin profiling followed protocols described in previous studies [14,15] with modifications. The mobile phase flow rate was at 0.4 mL min^−1^. In the mobile phase buffers, solvent A was water with 2% formic acid, and solvent B was 70% acetonitrile with 2% formic acid. The following gradient program was as follows: 0 to 5 min, 85% A and 15% B; 5 to 23 min, 70% A and 30% B; 23 to 30 min, 0% A and 100% B; and 30 to 35 min, 85% A and 15% B. The signal settings in the positive and negative mode were as follows: *m*/*z*, 100 to 1500; fragmentor, 70; and scan mode. The quantification was achieved using a calibration curve obtained for the cyanidin-3-O-glucoside standard.

### 2.5. Radical Scavenging Assays

The radical scavenging activities of DPPH (hydrophobic) and ABTS^+^ (both hydrophilic and phobic) reflect their capacity to eliminate reactive oxygen species (ROS) in lipid and aqueous environments, making them useful for investigating potential protective mechanisms against oxidative damages in various human diseases [16]. Superoxide scavenging activity is particularly important for neutralizing ROS produced in mitochondria, preventing DNA damage and cell death. Fe^2+^ chelation inhibits the Fenton reaction and helps alleviate metal-induced oxidative stress, which is strongly associated with diabetes and neurodegenerative diseases [17,18]. Reducing power, as measured by the FRAP assay, plays a role in maintaining intercellular redox balance, thereby supporting tissue protection and anti-inflammatory functions [19,20]. Each antioxidant assay thus reflects distinct pathological relevance and can serve as a basis for interpreting disease-specific antioxidant defense mechanisms. Therefore, we conducted various radical scavenging assays as described below.

For the DPPH radical scavenging assay, berry pre-extracts in methanol (Sigma-Aldrich, St. Louis, IL, USA) were diluted in 70% ethanol (Sigma-Aldrich, St. Louis, IL, USA) to five concentrations: 10, 30, 100, 300, and 1000 µg mL^−1^. A volume of 120 µL of each diluted extract was added to a 96-well microplate (Thermo Fisher Scientific Inc., Waltham, MA, USA) in triplicate. The diluted 70% ethanol with methanol served as the control. Next, 60 µL of prepared 0.45 mM DPPH reagent (Sigma-Aldrich, St. Louis, IL, USA) was added to each well. The samples were incubated in the dark for 15 min and absorbance at 517 nm was measured using a microplate reader (Multiskan SkyHigh, Thermo Scientific, Waltham, MA, USA).

For the ABTS^+^ radical scavenging assay, pre-extracts were diluted with deionized water to the same concentrations as in the DPPH assay. A volume of 100 µL of each diluted solution was mixed with 100 µL of ABTS^+^ reagent (G-Biosciences, St. Louis, MO, USA) in a 96-well microplate and reacted for 7 min. The ABTS^+^ was prepared by mixing 7 mM ABTS with 2.45 mM potassium persulfate and incubating the mixture in the dark for 16 h, following the manufacturer’s instructions. The ABTS^+^ solution had a base absorbance range of 0.7 to 0.8 at 734 nm, measured after the reaction using the microplate reader.

For the superoxide radical scavenging assay, pre-extracts were diluted using the following ABTS^+^ assay. Each well received 10 µL of diluted extracts, 40 µL of 0.1 M phosphate buffer (pH 7.8), 100 µL of a mixture containing 0.4 mM xanthine and 0.24 mM nitroblue tetrazolium (as spectrophotometric probes for superoxide anion radical), and 100 µL of 0.05 unit mL^−1^ xanthine oxidase (CheKine assay kit, MyBioSources, San Diego, CA, USA). Samples were incubated in the dark at 37 °C for 20 min, and absorbance was measured at 560 nm using the microplate reader.

The ferric-reducing activity power (FRAP) solution was prepared by mixing 300 mM of acetate buffer (pH 3.8), 10 mM of iron[III]-2,4,6-tripyridyl-*S*-triazine (TPTZ) solution, and 20 mM of ferric chloride (10:1:1, *v*/*v*/*v*). Pre-extracts were diluted in the acetate buffer to the same concentrations as in the DPPH assay. In a 96-well plate, 25 µL of diluted extract was reacted with 175 µL of FRAP solution (Sigma-Aldrich, St. Louis, IL, USA). The mixture was incubated in the dark for 10 min, and absorbance was measured at 593 nm using the microplate reader.

For the Fe^2+^ chelating assay kit (Zen-Bio Inc., Durham, NC, USA), pre-extracts were diluted in methanol to the same five concentrations (10, 30, 100, 300, and 1000 µg mL^−1^). A volume of 120 µL of diluted extract was reacted with 30 µL of 0.6 mM iron[II] chloride and 5 mM of bipyridyl in a 96-well microplate. After incubation in the dark for 10 min, absorbance was measured at 562 nm using the microplate reader.

Each antioxidant assay was conducted in triplicate. Radical scavenging activity was calculated using the following equation: radical scavenging capability (%) = (1−(A/B))×100, where A is the absorbance of the antioxidant compound reaction, and B is the absorbance of the control.

### 2.6. Statistical Analysis

Quantitative data were analyzed using analysis of variance with multiple comparisons in SPSS version 12 (SPSS Inc., Chicago, IL, USA). Pearson’s correlation coefficients (PCCs) were calculated at significance levels of *p* < 0.05 and *p* < 0.01, using the SPSS program. Correlation results were visualized using a red-to-blue gradient in Excel 2016 (Microsoft, Redmond, WA, USA) to represent the strength and direction of correlation.

## 3. Results

To evaluate antioxidant content and activity in *Rubus* species either cultivated or naturally grown in Korea, we collected a total of 15 genetic resources. Among them, three cultivars—V7, Maple, and Blackpearl—are *Rubus ursinus* varieties cultivated on farms and are commonly referred to as blackberries due to their darker fruit color (Table 1). In addition, three boysenberry cultivars (hybrids between *Rubus ursinus* and *R. idaeus*)—resources A, B, and C—also exhibited dark-colored fruits, although their pigmentation was less intense than that of the three blackberries (Table 1 and Figure 1). Among the wild raspberries, *Rubus crataegifolius*, *R. coreanus*, *R. parvifolius*, and *R. phoenicolasius* are locally known in Korea as “santtalgi”, “bokbunjattalgi”, “meongseokttalgi”, and “gomttalgi”, respectively. Except for *R. coreanus*, these species typically bear red-colored fruits (Table 1). Interestingly, within the raspberry group, *R. crataegifolius* was found to have elliptic leaves with lobed margins, while the others had ovate or elliptic leaf shapes (Table 1). These observations in fruit color and leaf morphology suggest possible differences in antioxidant composition and activity.

As a first step, we investigated the relationship between fruit color and chromaticity in the collected *Rubus* accessions. The colors of fruit extract varied depending on the genetic resource (Figure 1). According to Hunter’s color value, resources with high lightness value (>80) included no. 7, 9, 13, 14, and 15 (resources D, F, J, K, and N). These resources also showed lower redness, which was inversely proportional to lightness (Table 2). In general, blackberries, boysenberries, and bokbunjattalkies displayed higher redness values, while wild raspberries (santtalki, meongseokttalgi, and gomttalgi) exhibited lower redness (Table 2). Interestingly, yellowness varied among genetic resources independently of lightness and redness (Table 2). These results indicate that redness is associated with darker fruit coloration.

To determine whether TPC and TFC influence antioxidant activity and fruit color, both biochemical traits were measured. The highest TPC was observed in the resource F (no. 9), with 7.54 mg g^−1^, while the highest TFC was found in resource I (no. 12), with 7.52 mg g^−1^ (Table 3). Both were wild raspberry types. The top five resources for TPC were resource F, I, Maple, G, and V7 (no. 9, 12, 2, 10, and 1, respectively), and for TFC, they were resource I, G, F, Maple, and H (no. 12, 10, 9, 2, and 11, respectively) (Table 3). In our results, TFC levels were generally up to twice as high as TPC, although in resource F (no. 9), TPC was exceptionally higher than TFC and in other genetic resources (Table 3). Notably, black coloration did not consistently correspond to higher TPC and TFC. This suggests that certain pigments associated with black fruit color may be chemically distinct from the major antioxidant compounds.

To further assess the relationship between antioxidant content and activity, we analyzed the five types of antioxidant activity (DPPH radical scavenging, ABTS^+^ radical scavenging, superoxide radical scavenging, Fe^2+^ chelating, and RFAP). Extracts at five concentrations (µg mL^−1^) were reacted with each reagent, and the resulting activity (%) was calculated (Appendix A). From these data, 50% inhibition concentrations (IC_50_) were derived using scatter plots and linear regression in Excel (Appendix A and Table 4).

The top five resources with the lowest IC_50_ values, in order of performance, were as follows: for DPPH radical scavenging activity, resources F, V7, E, D, and G (no. 9, 1, 8, 7, and 10, respectively); for ABTS^+^ radical scavenging activity, resources F, V7, D, I, and C (no. 9, 1, 7, 12, and 6, respectively); for superoxide radical scavenging activity, resources J, Maple, H, C, and A (no. 13, 2, 11, 6, and 4, respectively); and for Fe^2+^ chelating activity, resources N, B, F, A, and C (no. 14, 5, 9, 4, and 6, respectively). Among these, resource F (no. 9) showed particularly strong activity in both the DPPH and ABTS^+^ assays (Table 4).

In the FRAP assay, resource F also demonstrated rapid saturation at 300 µg mL^−1^, indicating strong reducing power compared to other resources (Table 5). The top five resources for FRAP activity were F, V7, D, C, and Maple (no. 9, 1, 7, 6, and 2), which included three boysenberries as well as representatives of *R. crataegifolius* and *R. parvifolius* (Table 5 and Table 6).

All four scavenging activities as well as FRAP showed some degree of positive correlation between one another; in particular, ABTS^+^ and DPPH, Fe^2+^ chelating and superoxide, and ABTS^+^ and FRAP radical scavenging activities were significantly correlated (Figure 2). DPPH, ABTS^+^, and FRAP assay, which are used for detecting electron and hydrogen ion transfer mechanism in antioxidant activity [16,21], displayed similar top five rankings across resources (Table 3 and Table 6). Fe^2+^ chelating and FRAP, both involving metal chelation and electron transfer mechanisms, showed high activity in *R. parvifolius* and boysenberries (Table 6). In addition, TPC and TFC were significantly correlated with each other and negatively correlated with the IC_50_ of ABTS^+^ and FRAP activity (Figure 2). Since a lower IC_50_ value indicates higher antioxidant activity, this negative correlation supports the idea that higher levels of TPC and TFC enhance antioxidant activity [22]. Taken together, these results indicate that the most desirable resource, based on high scoring across analyses, is resource F (no. 9), as summarized in Table 6.

HPLC-MS analyses of each fruit extract were performed to investigate the specific compounds associated with coloration, antioxidant activity, and TPC and TFC, as potential functional food ingredients. Analysis of the 15 berry extracts revealed that all samples contained two major anthocyanin compounds, cyanidin-O-hexoside and cyanidin-3-O-glucoside (Figure 3), which were detected at retention times of 4.9 and 6.0 min, respectively, consistent with previous reports [14,23].

No other anthocyanin compounds were detected in this study (Figure 3A), indicating that coloration in *Rubus* berries is primarily associated with these two pigments. When we analyzed anthocyanin content, santtalki (*R. crataegifolius*), meongseokttalgi (*R. parvifolius*), and gomttalgi (*R. phoenicolasius*) fruits contained lower anthocyanin levels compared to blackberry (*R. ursinus*), boysenberry (hybrid *R. ursinus* and *idaeus*), and bokbunjattalki (*R. coreanus*) (Figure 3B). According to Table 1, the latter group exhibited black fruit coloration, while all the former displayed red coloration, suggesting that darker fruit color correlates with higher anthocyanin content.

The chromatographic profiles of 15 genetic resources could be divided into two groups: those with high anthocyanin content (more than 100 mg g^−1^ extract; no. 1 to 6, 10 to 12) and those with low content (less than 100 mg g^−1^ extract; no. 7 to 9, 13 to 15). Among the high-content group, resource no. 12 ranked first in TFC and second in TPC, corresponding to its higher anthocyanin levels (Figure 3 and Table 6). However, ranking alignment of anthocyanin levels did not perfectly match TFC and TPC. In addition, ranking alignments between anthocyanin levels and five antioxidant activities similarly showed poor correlation. Consistent with the Pearson correlation coefficient analysis, these results indicated that anthocyanins were not significantly associated with five radical scavenging assays (Figure 2). Only TFC significantly showed the highest correlation with anthocyanin compounds and content (Figure 2). Notably, within the low-content group, particularly resource no. 9 (*R. crataegifolius*), relatively high rankings were observed, with species in the top five for IC_50_ antioxidant activity (Figure 3 and Table 6). These findings suggest that antioxidant activity in *Rubus* fruits is not solely determined by anthocyanin content but also by other organic molecules that contribute to the observed activity across genetic resources.

## 4. Discussion

*Rubus* berries have garnered significant attention in the functional food market due to their diverse antioxidant compounds. These bioactive substances, naturally present in *Rubus* fruits, are known for their beneficial effects on human health and have been utilized in various applications for functional foods. In this study, we analyzed 15 *Rubus* genetic resources to determine which accessions exhibit superior bioactivity. We evaluated fruit coloration, TPC, TFC, anthocyanin composition, and five types of antioxidant activity in mature fruits. Fruit coloration was broadly categorized as red or black, and these variations were closely associated with anthocyanin content. In particular, cyanidin-O-hexoside and cyanidin-3-O-glucoside were identified as the predominant anthocyanins. Although *Rubus coreanus* and *R. ursinus* accessions showed relatively high TPC and TFC, the highest levels were observed in resource F (no. 9; *R. crataegifolius*), which also performed strongly in four antioxidant activity assays. These findings suggest that the selection of elite *Rubus* germplasms should not rely solely on anthocyanin content. Rather, a comprehensive evaluation involving multiple phytochemical indicators and antioxidant activities is essential to identify high-quality genetic resources suitable for breeding and functional food applications.

### 4.1. Coloration of Rubus Berries

The results of fruit color chromaticity and anthocyanin composition suggest that black coloration in *Rubus* berries is correlated with higher accumulation of cyanidin-O-hexoside and cyanidin-3-O-glucoside, particularly when fruits are darker in color (Figure 1 and Figure 3). This coloration is primarily due to the biosynthesis and accumulation of these anthocyanin pigments. Anthocyanin biosynthesis in *Rubus* fruit is regulated by specific structural genes such as *chalcone synthase*, *flavanone hydroxylase*, and *anthocyanidin synthase*, as well as transcription factors including *MYB*, *bHLH*, and *WD40* [24]. The upregulation of these genes during fruit ripening leads to enhanced production of cyanidin derivatives, contributing to the dark pigmentation. For example, *MYB10* expression in ripe fruit of blackberry (*R. occidentalis* L.) was higher than in other colored raspberries (*R. idaeus* L.; Jeltii gigant, Caroline, and American 22) [25]. Given the observed color variation among different accessions in this study, further molecular investigations would be valuable to elucidate the genetic basis. Moreover, the decrease in L* values (lightness) and increase in a* values (redness) observed in black-colored fruits (Figure 1 and Table 2) reflect the increased anthocyanin level (Figure 3). A study [26] reported a negative correlation between L* and anthocyanin content. Similarly, previous studies have shown that cyanidin-3-O-glucoside is a major contributor to purple and dark coloration in blackberries and mulberries [27] as well as in caneberry [28]. Therefore, higher anthocyanin levels are associated with the black coloration of *Rubus* fruits.

These differences in pigmentations have also been used as a phenotypic marker in breeding programs. *Rubus coreanus* (commonly known as bokbunjattalki), often misidentified as *R. occidentalis* [29], is widely recognized for its efficacy and used in traditional Korean raspberry wine [30]. Its black coloration distinguishes it from red-colored wild raspberries. Based on this trait, domestication and breeding efforts for bokbunjattalki have focused on genetic selection and trait enhancement, particularly for antioxidant capacity [31]. However, relying solely on black fruit coloration is not a comprehensive or reliable strategy for identifying superior *Rubus* varieties as functional foods. While black-colored fruits may appeal to consumer preferences from a marketing standpoint, this visual characteristic alone does not necessarily reflect the fruit’s bioactive potential. Therefore, it is essential to evaluate antioxidant activity alongside other phytochemical components to accurately assess and verify the health benefits of each accession.

### 4.2. Relationship Between Antioxidant Activity and Anthocyanin

All *Rubus* berries used in this study contained only two major anthocyanins, cyanidin-O-hexoside and cyanidin-3-O-glucoside (Figure 3), which primarily determine fruit coloration depending on pH conditions in fruit [32]. However, rankings based on anthocyanin content did not align with those of TPC, TFC, or antioxidant activities (Table 6). This discrepancy likely results from the distribution of other organic compounds such as phenolic acids (e.g., hydroxybenzoic acid and hydroxycinnamic acid), ellagitannins, vitamin C (ascorbic acid), and flavonoids (e.g., carotene, quercetin, and kaempferol) rather than anthocyanins alone [1,3,32]. For example, *Rubus* berries are rich in ellagitannins, which account for 51 to 88% of TPC and are key contributors to antioxidant activity [9,33]. The ellagic acid content in boysenberry also varied among its mutants and can reach levels similar to those in blackberry (V3) [14]. Furthermore, the ellagic acid content in raspberries can be affected by environmental growth conditions [34]. Fresh raspberries (*R. idaeus*) contain approximately 1.23 µmol g^−1^ (typically 0.05 to 0.4 mg g^−1^) of ascorbate, which is about three times higher than in blueberries [2,35]. Additionally, compounds such as hydroxybenzoic acid, hydroxycinnamic acid, and flavanols (quercetin and kaempferol) have been detected in *R. rosifolius* extracts, although in lower amounts than ellagic acid [36]. These studies and our results suggest that antioxidant activity in *Rubus* berries varies according to the distribution of multiple antioxidant compounds and environmental conditions.

### 4.3. Relationship Between Antioxidant Activity and Healthcare

Due to the reasons outlined above, *Rubus* berries have high market value and potential as functional foods rich in antioxidants [37]. *Rubus* fruits contain abundant phenolic compounds, anthocyanins, ellagitannins, vitamin C, and vitamin E [38]. These compounds in *Rubus* fruits may offer various health benefits, including reducing the risk of chronic diseases, inhibiting cancer cell proliferation, and modulating estrogenic activity [39,40,41]. Furthermore, short-term supplementation with *R. coreanus* Miquel enhanced antioxidant capacity in a healthy population [42]. In addition, clinical trials have shown that intake of unripe fruit (*R. coreanus*) significantly reduces cholesterol in patients with metabolic syndrome [43]. Daily consumption of red raspberries (*R. idaeus* L.) has also demonstrated that omics profiling in subjects at risk of metabolic syndrome showed effects on immune and signaling pathways [44]. These human-related or clinical research findings provide an important basis for evaluating *Rubus* berries as functional foods or natural therapeutics.

Based on antioxidant activity rankings and principle of radical scavenging assays, we can infer which genetic resources may be most effective for specific health applications. Resources J and K (*R. parvifolius*; no. 13 and 14, respectively) showed the highest superoxide and Fe^2+^ chelating activities (Table 6), suggesting strong potential for neuroprotection and diabetes prevention [45]. Resource no. 9, which ranked highly in the DPPH, ABTS^+^, and FRAP assays, may offer benefits related to LDL oxidation, atherosclerosis, and inflammation [18,19]. Therefore, accessions that perform strongly across all antioxidant activity assays hold promise for functional food market development, natural therapeutics, and breeding applications, although further clinical validation is needed.

A major strength of this study lies in its integrated evaluation of both wild and cultivated *Rubus* accessions collected across various regions of Korea. By combining chromaticity assessment, comprehensive phytochemical quantification, multiple antioxidant assays, and anthocyanin identification via HPLC-MS, we provide a systematic framework for identifying elite accessions. This approach enables direct comparison of germplasms under uniform analytical conditions, reducing methodological variability. However, this study also has limitations. The sampling was restricted to a single season and a limited number of geographical locations, which may not fully capture the annual or regional variability in phytochemical content. Moreover, the antioxidant activity assays were conducted exclusively in vitro, and thus the physiological relevance of these results in vivo remains to be further verified. Finally, other bioactive compounds potentially contributing to the antioxidant activity were not profiled. Thus, conducting long-term evaluations under diverse environmental and cultivation conditions in the future, together with in vivo validation and mechanistic studies of functional compounds, will enable the establishment of a precise resource selection framework applicable to the development of functional foods and plant breeding.

## 5. Conclusions

This study revealed that the antioxidant activity in *Rubus* berries is more closely related to TPC and TFC than to anthocyanin concentration or fruit color alone. Among the 15 accessions, a wild *R. crataegifolius* (no. 9, resource F) exhibited the highest antioxidant potential, indicating that red-colored berries can also serve as valuable functional resources. These findings suggest that the selection of superior *Rubus* germplasms for functional food applications or breeding should be guided by comprehensive phytochemical and bioactivity profiling rather than by visual traits such as fruit color.

From a practical perspective, the multiple antioxidant assays and anthocyanin characterization presented here can serve as selection criteria for both functional food industries and plant breeders aiming to develop high-value *Rubus* cultivars with enhanced health benefits. Future studies should validate these results in vivo, explore seasonal and environmental influences on phytochemical composition, and investigate the molecular mechanisms underlying the observed bioactivities. Furthermore, large-scale screening of germplasm resources and field trials would be valuable for breeding programs.

## Figures and Tables

**Figure 1 antioxidants-14-01012-f001:**
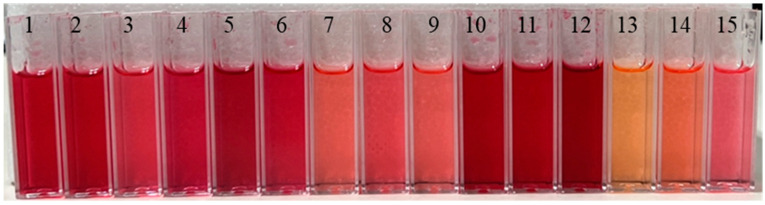
Coloration of berry extracts from *Rubus* genus species. The berries were extracted using methanol. The numbers in the photo correspond to the genetic resources listed in Table 1. 1, V7; 2, Maple; 3, Blackpearl; 4 to 13, Resource A to J, respectively; 14, Resource K; 15, Resource N.

**Figure 2 antioxidants-14-01012-f002:**
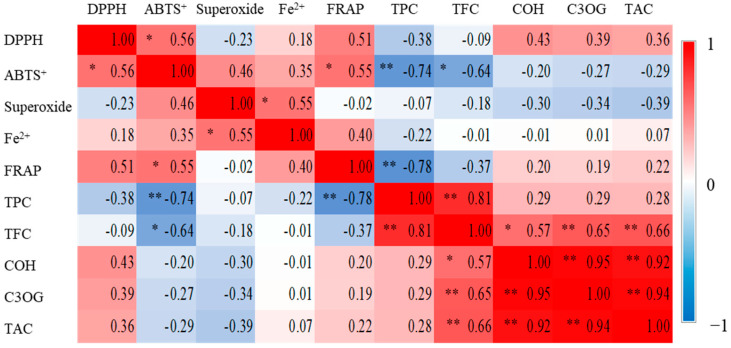
Pearson correlation coefficient (PCC) analysis among IC_50_ antioxidant activities, total phenolic content (TPC), total flavonoid content (TFC), and anthocyanins. PCC values close to red and blue represent positive and negative correlations, respectively. Asterisks indicate statistical significance; ** *p* < 0.01; * *p* < 0.05. FRAP, ferric-reducing activity power; COH, cyanidin-O-hexoside; C3OG, cyanidin-3-O-glucoside, TAC, total anthocyanin content.

**Figure 3 antioxidants-14-01012-f003:**
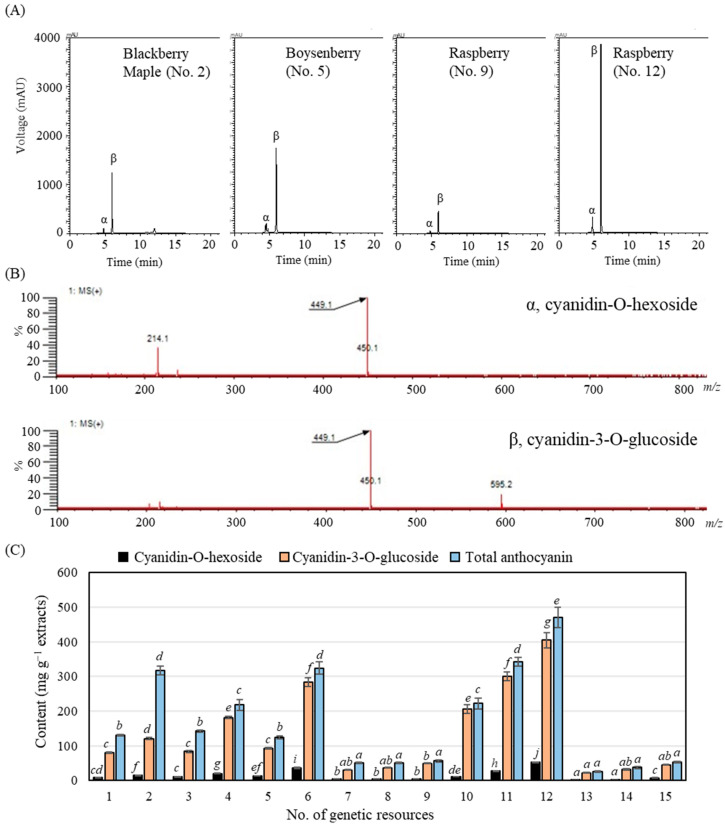
Chromatographic profile (**A**), mass spectrometry (**B**), and content (**C**) of anthocyanin compounds in *Rubus* accessions. (**A**) Chromatographic profiles of anthocyanins in genetic resource no. 2, 5, 9, and 12. The two major anthocyanin compounds are α (cyanidin-O-hexoside) and β (cyanidin-3-O-glucoside), which were known as properties of *Rubus* accessions found in Korea [15]. Both compounds have 449 *m*/*z* (M^+^). The λ_max_ of α and β was 517 and 519 nm, respectively. (**C**) Anthocyanin content in berry extracts from 15 genetic resources. Calibration curve used for quantification: cyanidin-3-O-glucoside (*y* = 20,601*x* − 523.43, *R*^2^ = 0.9999). Different letters in each compound indicate statistically significant differences among resources based on Tukey’s test (*p* < 0.05).

**Table 1 antioxidants-14-01012-t001:** List of berry genetic resources used in this study.

No.	Resources	Common Name	Korean Name	Botanical Name (Scientific Name)	Origin of Resource	Donor	Extraction Solvent for Anthocyanin	Fruit Shape and Color	Leaf Shape
1	V7	Blackberry	-	*Rubus ursinus*	Wanju, Jeonbuk, Korea	Hanjik Cho	1% formic acid in absolute methanol	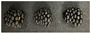	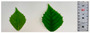
2	Maple	Wanju, Jeonbuk, Korea	Hanjik Cho	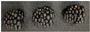	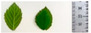
3	Blackpearl	Wanju, Jeonbuk, Korea	Hanjik Cho	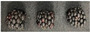	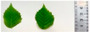
4	Resource A	Boysenberry	-	*Rubus ursinus* × *Rubus idaeus*	Wanju, Jeonbuk, Korea	Hanjik Cho	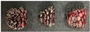	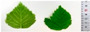
5	Resource B	Wanju, Jeonbuk, Korea	Hanjik Cho	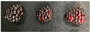	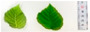
6	Resource C	USA to Wanju, Jeonbuk, Korea	Hanjik Cho	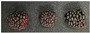	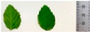
7	Resource D	Raspberry	Santtalki	*Rubus crataegifolius*	Jeongeup, Jeonbuk, Korea	Wild variety	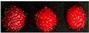	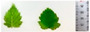
8	Resource E	Namwon, Jeonbuk, Korea	Wild variety	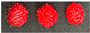	Not available
9	Resource F	Wanju, Jeonbuk, Korea	Wild variety	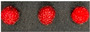	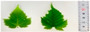
10	Resource G	Bokbunjattalki	*Rubus coreanus*	Jeongeup, Jeonbuk, Korea	Wild variety	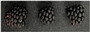	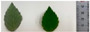
11	Resource H	Wanju, Jeonbuk, Korea	Wild variety	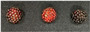	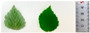
12	Resource I	Wanju, Jeonbuk, Korea	Hanjik Cho	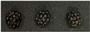	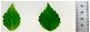
13	Resource J	Meongseokttalgi	*Rubus parvifolius*	Jeongeup, Jeonbuk, Korea	Wild variety	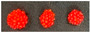	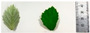
14	Resource K	Wanju, Jeonbuk, Korea	Wild variety	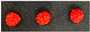	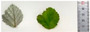
15	Resource N	Gomttalgi	*Rubus phoenicolasius*	Jeongeup, Jeonbuk, Korea	Wild variety	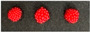	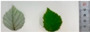

**Table 2 antioxidants-14-01012-t002:** Color value in berry extracts from *Rubus* species.

No.	Resources	Botanical Name (Scientific Name)	Hunter’s Color Value		
			Lightness (L*)	Redness (a*)	Yellowness (b*)
1	V7	*Rubus ursinus*	56.39 ± 0.01 ^i^	66.16 ± 3.76 ^a^	21.69 ± 2.28 ^d,e^
2	Maple		56.30 ± 0.39 ^i^	66.57 ± 0.80 ^a^	25.62 ± 3.59 ^c,d^
3	Blackpearl		71.74 ± 0.33 ^g^	45.17 ± 1.61 ^b^	9.21 ± 1.05 ^g,h^
4	Resource A	*Rubus ursinus* × *Rubus idaeus*	59.80 ± 0.60 ^h^	68.46 ± 1.75 ^a^	4.72 ± 0.53 ^h^
5	Resource B		54.64 ± 0.15 ^j^	68.56 ± 1.28 ^a^	12.41 ± 1.35 ^f,g,h^
6	Resource C		59.05 ± 0.38 ^h^	68.68 ± 0.86 ^a^	6.54 ± 3.11 ^g,h^
7	Resource D	*Rubus crataegifolius*	81.74 ± 0.29 ^e^	24.59 ± 2.27 ^d,e^	13.31 ± 1.47 ^e,f,g^
8	Resource E		78.48 ± 0.80 ^f^	33.82 ± 1.23 ^c^	9.92 ± 1.41 ^g,h^
9	Resource F		82.18 ± 0.16 ^c,d^	26.78 ± 1.28 ^d^	12.70 ± 1.70 ^f,g,h^
10	Resource G	*Rubus coreanus*	50.52 ± 0.44 ^l^	68.13 ± 2.32 ^a^	43.44 ±2.48 ^b^
11	Resource H		52.70 ± 0.72 ^k^	68.91 ± 0.72 ^a^	33.94 ± 2.85 ^c^
12	Resource I		47.96 ± 0.36 ^m^	67.71 ± 0.71 ^a^	53.73 ± 7.19 ^a^
13	Resource J	*Rubus parvifolius*	88.61 ± 0.35 ^a^	9.88 ± 0.71 ^f^	26.78 ± 1.96 ^c,d^
14	Resource K		84.63 ± 0.37 ^b^	20.67 ± 0.84 ^e^	20.62 ± 2.28 ^d,e,f^
15	Resource N	*Rubus phoenicolasius*	83.12 ± 0.61 ^c^	26.91 ± 2.00 ^d^	6.19 ± 2.05 ^g,h^

Values denote mean ± standard deviation (*n* = 3). Different superscripts in the same column indicate significant differences based on Tukey’s test (*p* < 0.05).

**Table 3 antioxidants-14-01012-t003:** Total polyphenol and flavonoid content in various *Rubus* berries.

No.	Resources	Botanical Name (Scientific Name)	Total Phenolic Contents (TAE, mg g^−1^ of Frozen Fruits)	Total Flavonoid Content (QE, mg g^−1^ of Frozen Fruits)
1	V7	*Rubus ursinus*	3.40 ± 0.09 ^c,d^	4.54 ± 0.17 ^e,f^
2	Maple		3.56 ± 0.04 ^c^	5.99 ± 0.28 ^c^
3	Blackpearl		1.92 ± 0.01 ^g^	3.72 ± 0.08 ^g^
4	Resource A	*Rubus ursinus* × *Rubus idaeus*	3.18 ± 0.10 ^d,e^	5.06 ± 0.11 ^d,e^
5	Resource B		3.36 ± 0.01 ^c,d^	4.82 ± 0.14 ^e^
6	Resource C		3.34 ± 0.21 ^c,d^	5.11 ± 0.16 ^d,e^
7	Resource D	*Rubus crataegifolius*	3.05 ± 0.06 ^e^	4.43 ± 0.15 ^e,f^
8	Resource E		2.51 ± 0.02 ^f^	4.16 ± 0.21 ^f,g^
9	Resource F		7.54 ± 0.10 ^a^	6.90 ± 0.25 ^a,b^
10	Resource G	*Rubus coreanus*	3.46 ± 0.03 ^c,d^	7.02 ± 0.38 ^a,b^
11	Resource H		3.34 ± 0.08 ^c,d^	5.76 ± 0.20 ^c^
12	Resource I		4.85 ± 0.20 ^b^	7.52 ± 0.14 ^a^
13	Resource J	*Rubus parvifolius*	2.52 ± 0.03 ^f^	4.20 ± 0.25 ^f,g^
14	Resource K		1.03 ± 0.04 ^i^	2.75 ± 0.18 ^h^
15	Resource N	*Rubus phoenicolasius*	1.43 ± 0.02 ^h^	3.22 ± 0.14 ^g,h^

Values denote mean ± standard deviation (*n* = 3). Different superscripts in the same column indicate significant differences based on Tukey’s test (*p* < 0.05). TAE, tannic acid equivalent; QE, quercetin equivalent.

**Table 4 antioxidants-14-01012-t004:** Antioxidant activity in various *Rubus* berries.

No.	Resources	Botanical Name (Scientific Name)	DPPH Radical * IC_50_ (µg mL^−1^)	ABTS^+^ Radical IC_50_ (µg mL^−1^)	Superoxide Radical IC_50_ (µg mL^−1^)	Fe^2+^ Chelating IC_50_ (µg mL^−1^)
1	V7	*Rubus ursinus*	112.15 ^#2^	59.74 ^#2^	201.75	415.68
2	Maple		240.82	73.31	155.66 ^#2^	507.28
3	Blackpearl		237.41	97.11	213.64	459.56
4	Resource A	*Rubus ursinus* × *Rubus idaeus*	276.58	84.42	189.89 ^#5^	296.11 ^#4^
5	Resource B		246.25	111.54	245.51	263.74 ^#2^
6	Resource C		212.61	68.93 ^#5^	173.87 ^#4^	343.98 ^#5^
7	Resource D	*Rubus crataegifolius*	169.52 ^#4^	60.52 ^#3^	296.62	509.39
8	Resource E		156.26 ^#3^	156.26	706.60	707.53
9	Resource F		45.88 ^#1^	8.53 ^#1^	272.33	266.40 ^#3^
10	Resource G	*Rubus coreanus*	204.40 ^#5^	72.13	246.67	457.14
11	Resource H		374.85	76.31	158.94 ^#3^	526.22
12	Resource I		421.09	65.55 ^#4^	192.40	457.24
13	Resource J	*Rubus parvifolius*	224.61	74.84	149.55 ^#1^	442.16
14	Resource K		225.83	111.78	210.61	238.87 ^#1^
15	Resource N	*Rubus phoenicolasius*	548.39	186.37	267.01	487.49

Sample concentrations tested for each assay were 10, 30, 100, 300, and 1000 µg mL^−1^. From the results of the scavenging assay shown in Appendix A, IC_50_ values are calculated based on concentration–response curves using mean values of each concentration. * 50% inhibitory concentration; ^#^ top 5 genetic resources in each assay (the numbers indicate the activity ranking, from highest to lowest); DPPH, 2-2-diphenyl-1-picrylhydrazyl; ABTS^+^, 2,2′-azino-bis(3-ethylbenzothiazoline-6-sulfonic acid) cation.

**Table 5 antioxidants-14-01012-t005:** Ferric-reducing activity power in various *Rubus* berries.

No.	Resources	Botanical Name (Scientific Name)	Ferric-Reducing Activity Power (O.D = 593 nm)				
			Concentrations (µg mL^−1^)				
			10	30	100	300	1000
1	V7	*Rubus ursinus*	0.053 ± 0.001 ^g^	0.137 ± 0.000 ^fg^	0.390 ± 0.008 ^f^	1.024 ± 0.018 ^h^	^h^ 2.392 ± 0.065 ^#3^
2	Maple		0.049 ± 0.001 ^fg^	0.113 ± 0.002 ^g^	0.321 ± 0.007	0.847 ± 0.010 ^fg^	^fg^ 2.187 ± 0.052
3	Blackpearl		0.039 ± 0.002 ^cd^	0.082 ± 0.001 ^abc^	0.231 ± 0.002 ^c^	0.641 ± 0.021 ^bcd^	^de^ 1.787 ± 0.047
4	Resource A	*Rubus ursinus* × *Rubus idaeus*	0.040 ± 0.003 ^cde^	0.093 ± 0.002 ^def^	0.292 ± 0.002 ^d^	0.765 ± 0.002 ^efg^	^f^ 2.136 ± 0.051
5	Resource B		0.037 ± 0.001 ^c^	0.080 ± 0.001 ^bcd^	0.218 ± 0.007 ^c^	0.582 ± 0.009 ^bc^	^cd^ 1.682 ± 0.037
6	Resource C		0.044 ± 0.002 ^def^	0.110 ± 0.002 ^f^	0.330 ± 0.007 ^e^	0.883 ± 0.005 ^fg^	^gh^ 2.286 ± 0.037 ^#4^
7	Resource D	*Rubus crataegifolius*	0.045 ± 0.002 ^ef^	0.106 ± 0.002 ^ef^	0.335 ± 0.004 ^e^	0.922 ± 0.015 ^g^	^i^ 2.581 ± 0.056 ^#2^
8	Resource E		0.037 ± 0.002 ^c^	0.081 ± 0.001 ^bcde^	0.238 ± 0.002 ^c^	0.655 ± 0.017 ^cde^	^e^ 1.838 ± 0.047
9	Resource F		0.125 ± 0.001 ^h^	0.326 ± 0.007 ^h^	0.985 ± 0.024 ^g^	2.527 ± 0.011 ^i^	^j^ 2.940 ± 0.005 ^#1^
10	Resource G	*Rubus coreanus*	0.042 ± 0.001 ^cde^	0.090 ± 0.001 ^cdef^	0.271 ± 0.016 ^d^	0.739 ± 0.027 ^cde^	^f^ 2.103 ± 0.029
11	Resource H		0.038 ± 0.004 ^c^	0.095 ± 0.001 ^def^	0.270 ± 0.000 ^d^	0.727 ± 0.032 ^cde^	^f^ 2.094 ± 0.029
12	Resource I		0.039 ± 0.002 ^cd^	0.101 ± 0.004 ^def^	0.296 ± 0.007 ^de^	0.802 ± 0.018 ^efg^	^g^ 2.264 ± 0.018 ^#5^
13	Resource J	*Rubus parvifolius*	0.040 ± 0.001 ^cde^	0.081 ± 0.003 ^bcde^	0.233 ± 0.004 ^c^	0.606 ± 0.007 ^bcd^	^c^ 1.623 ± 0.029
14	Resource K		0.028 ± 0.001 ^b^	0.060 ± 0.002 ^ab^	0.170 ± 0.003 ^b^	0.466 ± 0.004 ^ab^	^b^ 1.133 ± 0.011
15	Resource N	*Rubus phoenicolasius*	0.022 ± 0.001 ^a^	0.042 ± 0.001 ^a^	0.110 ± 0.001 ^a^	0.294 ± 0.001 ^a^	^a^ 0.858 ± 0.008

Values denote mean ± standard deviation (*n* = 3). Different superscripts in the same column indicate significant differences based on Tukey’s test (*p* < 0.05). ^#^ top 5 genetic resources (the numbers indicate the activity ranking, from highest to lowest).

**Table 6 antioxidants-14-01012-t006:** Summary of the top 5 *Rubus* accessions for each antioxidant activity.

No.	Resources	Botanical Name (Scientific Name)	Order in Terms of Various Antioxidant Activity							
			TPC	TFC	DPPH Radical IC_50_	ABTS^+^ Radical IC_50_	Superoxide Radical IC_50_	Fe^2+^ Chelating IC_50_	FRAP	Score
1	V7	*Rubus ursinus*	5		2	2			2	4 ea
2	Maple		3	4			2		5	4 ea
3	Blackpearl									-
4	Resource A	*Rubus ursinus* × *Rubus idaeus*					5	4		2 ea
5	Resource B							2		1 ea
6	Resource C					5	4	5	4	4 ea
7	Resource D	*Rubus crataegifolius*			4	3			3	3 ea
8	Resource E				3					1 ea
9	Resource F		1	3	1	1		3	1	6 ea
10	Resource G	*Rubus coreanus*	4	2	5					3 ea
11	Resource H			5			3			2 ea
12	Resource I		2	1		4				3 ea
13	Resource J	*Rubus parvifolius*					1			1 ea
14	Resource K							1		1 ea
15	Resource N	*Rubus phoenicolasius*								-

Top five data points are taken from Table 3, Table 4 and Table 5. TPC, total phenolic content; TFC, total flavonoid content; FRAP, ferric-reducing activity power.

## Data Availability

All data generated or analyzed during this study are included in this article (and its Appendix A).

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
