# Peer review of "Evaluation of Anthocyanin Profiling, Total Phenolic and Flavonoid Content, and Antioxidant Activity of Korean Rubus Accessions for Functional Food Applications and Breeding"

_antioxidants, 2025, doi:10.3390/antiox14081012_

Round 1
Reviewer 1 Report
From my point of view, the manuscript submitted to Antioxidants can be considered for publication after the authors make some revisions. These are my suggestions:
The abstract needs to be properly revised: After a background statement and the justification to perform this research, the authors have to clearly state the study’s aims. The applied methods are not described, and the results are not expressed quantitatively with the statistical significance. Future perspectives and practical implications have to be elaborated after the conclusions.
The keyword “Rubus genus” should be replaced by “Rubus”.
The Introduction is too brief, and it doesn’t give the needed international overview on the addressed topics. In this section, the authors have to justify well the relevance of the study. The study’s objectives are not clarified.
The sections and subsections are not numbered.
The images of fruit shape and color (Table 1) have to be increased.
More details on the analysis of total phenolic and flavonoid content could be provided.
The quality and size of Figure 3 should be improved.
The Results and Discussion sections are adequate. My only advice is to include and discuss the study’s main strengths and limitations.
The Conclusions can be elaborated, and some directions for further studies as well as the practical implications should be pointed out.
From my point of view, the manuscript submitted to Antioxidants can be considered for publication after the authors make some revisions. These are my suggestions:
The abstract needs to be properly revised: After a background statement and the justification to perform this research, the authors have to clearly state the study’s aims. The applied methods are not described, and the results are not expressed quantitatively with the statistical significance. Future perspectives and practical implications have to be elaborated after the conclusions.
The keyword “Rubus genus” should be replaced by “Rubus”.
The Introduction is too brief, and it doesn’t give the needed international overview on the addressed topics. In this section, the authors have to justify well the relevance of the study. The study’s objectives are not clarified.
The sections and subsections are not numbered.
The images of fruit shape and color (Table 1) have to be increased.
More details on the analysis of total phenolic and flavonoid content could be provided.
The quality and size of Figure 3 should be improved.
The Results and Discussion sections are adequate. My only advice is to include and discuss the study’s main strengths and limitations.
The Conclusions can be elaborated, and some directions for further studies as well as the practical implications should be pointed out.
Author Response
Reviewer 1
General comment: From my point of view, the manuscript submitted to Antioxidants can be considered for publication after the authors make some revisions.
Response: We sincerely thank the reviewer for the positive overall evaluation of our manuscript and for recognizing its potential for publication in Antioxidants. We appreciate your constructive feedback and suggestions to improve the quality, clarity, and scientific rigor of the manuscript. We have carefully addressed all of your comments. Detailed responses to each point are provided below and in the revised version.
These are my suggestions:
Comment 1: The abstract needs to be properly revised: After a background statement and the justification to perform this research, the authors have to clearly state the study’s aims. The applied methods are not described, and the results are not expressed quantitatively with the statistical significance (in the abstract).
Response 1: Thank you for your constructive feedback. We have revised the Abstract following your suggestions as below, including justification, study’s aims, applied methods, quantitative expression, and statistical significance.
The revised Abstract now appears at line 15 to 32 in revised manuscript;
“The Rubus genus includes numerous berry species known for their rich phytochemical content and antioxidant properties. However, comparative evaluations of wild and cultivated Rubus germplasms in East Asia remain limited. This study aimed to identify superior resources with potential for use in functional foods and breeding through integrated phytochemical and antioxidant profiling. Fifteen accessions collected across Korea were assessed for fruit coloration, total phenolic content (TPC), total flavonoid content (TFC), five antioxidant activities (DPPH, ABTS+, superoxide, FRAP, and Fe2+ chelation), and anthocyanin composition by HPLC-MS. The TPC ranged from 1.03 to 7.54 mg g-1 of frozen fruit, and TFC ranged from 2.75 to 7.52 mg g-1 of frozen fruit, with significant differences among accessions (p < 0.05). Black-colored fruits such as R. coreanus and R. ursinus varieties exhibited high anthocyanin levels (approximately total 471 and 316 mg g-1 extracts, respectively), with cyanidin-O-hexoside and cyanidin-3-O-glucoside being the dominant pigments. However, the antioxidant performance of these accessions varied. A wild Rubus crataegifolius (no.9, resource F) showed the highest TPC and ranked within the top five in multiple antioxidant assays, despite its moderate anthocyanin content. Correlation analysis revealed that TPC and TFC were significantly associated with antioxidant activity (p < 0.05), but not directly with anthocyanin content. These results suggest that antioxidant potential is influenced by a broader spectrum of phenolic compounds, rather than anthocyanins alone. These findings underscore the need to look beyond visual traits and focus on biochemical evidence when selecting elite Rubus accessions.”
Comment 2: The keyword “Rubus genus” should be replaced by “Rubus”.
Response 2: Thank you for your comment. We have replaced the keyword in the revised version, at line 33.
Comment 3: The Introduction is too brief, and it doesn’t give the needed international overview on the addressed topics. In this section, the authors have to justify well the relevance of the study. The study’s objectives are not clarified.
Response 3: Thank you for your valuable suggestion. We have revised the final paragraph of Introduction to more clearly emphasize the novelty of our study. This change is reflected in the revised manuscript, and this now appears as below.
At line 69 to 78: “Despite the growing interest in Rubus fruit research, comparative studies involving both wild and cultivated Rubus germplasms are still limited, particularly in East Asia. In addition, bioactive compounds responsible for antioxidant properties in Rubus fruits also closely related to therapeutic or pharmacological
properties. These include an-ti-inflammatory, antimicrobial, anti-obesity, anticancer, and antidiabetic activities, which have been validated in both in vitro and in vivo models [5,10,11]. Such recent findings have elevated Rubus species as promising candidates not only functional food develop-ment but also for considerable pharmacological applications. Therefore, a deeper under-standing of their phytochemical and antioxidant properties is crucial for selecting elite accessions with superior biofunctional potential.”
At line 86 to 88: “To our knowledge, this is one of the few studies that systematically compares both wild and cultivated Rubus accessions from Korea.”
Comment 4: The sections and subsections are not numbered.
Response 4: Thank you for your comment. We have added numbered section and subsections as following author guidelines.
Comment 5: The images of fruit shape and color (Table 1) have to be increased.
Response 5: Thank you for your valuable feedback. We have revised Table 1 to improve the clarity of the fruit images, adjusting the layout and size to enhance visibility. However, we acknowledge that the quality of the original photographs (low resolution via cellphone camera) was limited. Despite our efforts to enhance image sharpness, some limitations remain due to the initial image resolution at the time of sample collection. We will appreciate your understanding and consider acquiring high-resolution images in future studies.
Comment 6: More details on the analysis of total phenolic and flavonoid content could be provided.
Response 6: Thank you for your valuable feedback. We have revised Material and Method section on the total phenolic and flavonoid content. This includes statement of standard curve, some words, and equation for quantity. The revised manuscript now appears as below:
At line 119 to 121: “Tannic acid (Sigma-Aldrich) was used to generate a standard curve as a standard (0 to 1 mg mL−1, R2 > 0.99) and the results were expressed as tannic acid equivalents (TAE, mg g-1).”
At line 127 to 131: “Quercetin (Sigma-Aldrich) was used to generate a standard curve as a standard (0 to 1 mg mL−1, R2 > 0.99). The results were expressed as quercetin equivalents (QE, mg g-1) using the following equation: TPC or TFC = C × V × M−1, where C is the standard (tannic acid or quercetin) concentration from the calibration curve, V is the extract volume, and M is the sample mass.”
Some words: Line 119, “against a methanol blank”; Line 123, “160μL of 60% (v/v) ethanol”
Comment 7: The quality and size of Figure 3 should be improved.
Response 7: Thank you for the helpful suggestion. In the revised manuscript, we have restructured Figure 3 to improve the image quality.
Comment 8: The Results and Discussion sections are adequate. My only advice is to include and discuss the study’s main strengths and limitations. The Conclusions can be elaborated, and some directions for further studies as well as the practical implications should be pointed out. Future perspectives and practical implications have to be elaborated after the conclusions.
Response 8: Thank you for your helpful suggestions. In the Discussion section, we have added explicit statements outlining the main strengths of this study and the main limitations. In the Conclusion section, we have elaborated on provided clear directions for future research and emphasized the practical implications. These revisions now appear in the revised manuscript, and below:
At line 433 to 448: “A major strength of this study lies in its integrated evaluation of both wild and culti-vated Rubus accessions collected across various regions of Korea. By combining chroma-ticity assessment, comprehensive phytochemical quantification, multiple antioxidant as-says, and anthocyanin identification via HPLC-MS, we provide a systematic framework for identifying elite accessions. This approach enables direct comparison of germplasms under uniform analytical conditions, reducing methodological variability. However, this study also has limitations. The sampling restricted to a single season and a limited num-ber of geographical locations, which may not fully capture the annual or regional variabil-ity in phytochemical content. Moreover, the antioxidant activity assays were conducted exclusively in vitro, and thus the physiological relevance of these results in vivo remains to be further verified. Finally, other bioactive compounds potentially contributing to the antioxidant activity were not profiled. Thus, conducting long-term evaluations under di-verse environmental and cultivation conditions in the future, together with in vivo valida-tion and mechanistic studies of functional compounds, will enable the establishment of a precise resource selection framework applicable to the development of functional foods and plant breeding.”
At line 457 to 463: “From a practical perspective, the multiple antioxidant assays and anthocyanin char-acterization presented here can serve as selection criteria for both functional food indus-tries and plant breeders aiming to develop high-value Rubus cultivars with enhanced health benefits. Future studies should validate these results in vivo, explore seasonal and environmental influences on phytochemical composition, and investigate the molecular mechanisms underlying the observed bioactivities. Furthermore, large scale screening of germplasm resources and field trials would be valuable for breeding programs.”

Reviewer 2 Report
The fruits of the Rubus genus are rich in total phenolic content (TPC) and total flavonoid content (TFC), which are widely utilized in functional foods. This manuscript reports the TPC and TFC levels and evaluates their antioxidant activities. The research topic is both scientifically interesting and aligned with the aims and scope of the journal.
1. The number and rationale for replicate testing should be clearly specified in the METHODS section.
2. In Lines 321–353, the underlying mechanism of Rubus berry coloration should be elaborated, rather than merely reiterating the results.
3. In Lines 354–379, content unrelated to Rubus berries should be replaced with a review of the antioxidant and health-promoting effects of Rubus berries on humans.
4. Lines 389–401 can be integrated into the section titled “Coloration of Rubus Berries” for improved coherence and organization.
5. The list of references must comply with the specific guidelines provided by the journal.
- The title should be revised to include TPC and TFC, rather than focusing solely on anthocyanin.
Author Response
Reviewer 2
General comment: The fruits of the Rubus genus are rich in total phenolic content (TPC) and total flavonoid content (TFC), which are widely utilized in functional foods. This manuscript reports the TPC and TFC levels and evaluates their antioxidant activities. The research topic is both scientifically interesting and aligned with the aims and scope of the journal.
Response: We sincerely thank the reviewer for the positive comments. The detailed responses to the reviewer’s comments and suggestions are provide below.
Comment 1: The number and rationale for replicate testing should be clearly specified in the METHODS section.
Response 1: Thank you for your comment. We have revised the Materials and Method section to clearly specify the number of replicates and the rationale for replicate. Accordingly, following statements at the Materials and Method section are now appear as below:
At line 96 to 99: “All measurements described below were performed in triplicate (n=3) to ensure statistical reliability and minimize experimental variability. This level of replication for phytochemical and antioxidant assays in Rubus fruits provides sufficient statistical power to detect significant differences among accessions.”
At line 112 to 113: “All TPC and TFC measurements were performed in triplicate to ensure reproducibility.”
At line 198: “Each antioxidant assay was conducted in triplicate.”
At line 150 to 162: We have restructured “Radical scavenging assays” in the Material and Method section as following comment 3 to give rationale for various assays. “The radical scavenging activities of DPPH (hydrophobic) and ABTS+ (both hydrophilic and phobic) reflect their capacity to eliminate reactive oxygen species (ROS) in lipid and aqueous environments, making them useful for investigating potential protective mechanisms against oxidative damages in various human diseases [16]. Superoxide scavenging activity is particularly important for neutralizing ROS produced in mitochondria, preventing DNA damage and cell death. Fe2+ chelation inhibits the Fenton reaction and helps alleviate metal-induced oxidative stress, which is strongly associated with diabetes and neurodegenerative diseases. [17,18]. Reducing power, as measured by the FRAP assay, plays a role in maintaining intercellular redox balance, thereby supporting tissue protection and anti-inflammatory functions [19,20]. Each antioxidant assay thus reflects distinct pathological relevance and can serve as a basis for interpreting disease-specific antioxidant defense mechanisms. Therefore, we conducted various radical scavenging assays as described below.”
Comment 2: In Lines 321–353, the underlying mechanism of Rubus berry coloration should be elaborated, rather than merely reiterating the results.
Response 2: Thank you for your valuable suggestion. We have elaborated on the underlying mechanism of Rubus fruit coloration in the revised manuscript. We have described key structural genes and transcription factors in anthocyanin biosynthesis to explain mechanism. The added references and citations have also revised as following author guidelines. This version now appears as like below:
At line 361 to 370: “This coloration is primarily due to the biosynthesis and accumulation of these anthocyanin pigments. Anthocyanin biosynthesis in Rubus fruit is regulated by specific structural genes such as chalcone synthase, flavanone hydroxylase, and anthocyanidin synthase, as well as transcription factors including MYB, bHLH, and WD40 [24]. The upregulation of these genes during fruit ripening leads to enhanced production of cyanidin derivatives, contributing to the dark pigmentation. For example, MYB10 expression in ripen fruit of black berry (R. occidentalis L.) was higher than in other colored raspberries (R. idaeus L.; Jeltii gigant, Caroline, and American 22)[25]. Given the observed color variation among different accessions in this study, further molecular investigations would be valuable to elucidate the genetic basis.”
Comment 3: In Lines 354–379, content unrelated to Rubus berries should be replaced with a review of the antioxidant and health-promoting effects of Rubus berries on humans.
Response 3: Thank you for your suggestion. We have moved statement for principles of radical scavenging assays into Material and Method section and replaced with a review of the antioxidant and health-promoting effects of Rubus berries on humans. The revised versions now appear instead, as below:
At line 411 to 422: “Rubus fruits contain abundant phenolic compounds, anthocyanins, ellagitannins, vitamin C, and vitamin E [38]. These compounds in Rubus fruits may offer various health benefits, including reducing the risk of chronic diseases, inhibiting cancer cell proliferation, and modulating estrogenic activity [39,40,41]. Furthermore, short-term supplementation with R. coreanus Miquel enhanced antioxidant capacity in a healthy population [42]. In addition, clinical trials have shown that intake of unripe fruit (R. coreanus) significantly reduces cholesterol in patients with metabolic syndrome [43]. Daily consumption of red raspberries (R. idaeus L.) has also demonstrated that omics profiling in subjects at risk of metabolic syndrome showed effects on immune and signaling pathways [44]. These human-related or clinical research findings provide an important basis for evaluating Rubus berries as functional foods or natural therapeutics.”
Comment 4: Lines 389–401 can be integrated into the section titled “Coloration of Rubus Berries” for improved coherence and organization.
Response 4: Thank you for your suggestion. Following your suggestion, the content in lines 389-401 has been integrated into the section titled “Coloration of Rubus berries”. During this process, we revised some expressions and citations to ensure a smooth connection with next section.
At line 377 to 388: “These differences in pigmentations have also been used as a phenotypic marker in breeding programs. Rubus coreanus (commonly known as bokbunjattalki), often misidenti-fied as R. occidentalis [29], is widely recognized for its efficacy and used in traditional Ko-rean raspberry wine [30]. Its black coloration distinguishes it from red-colored wild rasp-berries. Based on this trait, domestication and breeding efforts for bokbunjattalki have fo-cused on genetic selection and trait enhancement, particularly for antioxidant capacity [31]. However, relying solely on black fruit coloration is not a comprehensive or reliable strategy for identifying superior Rubus varieties as functional foods. While black-colored fruits may appeal to consumer preferences from a marketing standpoint, this visual char-acteristic alone dose not necessarily reflect the fruit’s bioactive potential. Therefore, it is essential to evaluate antioxidant activity alongside other phytochemical components to accurately assess and verify the health benefits of each accession.”
Comment 5: The list of references must comply with the specific guidelines provided by the journal.
Response 5: Thank you for pointing this out. We have carefully reviewed the entire manuscript and removed duplicated reference format. All in-text citations and reference styles now appear in the appropriate numbered format only, as follow the journal guidelines.
Comment 6. The title should be revised to include TPC and TFC, rather than focusing solely on anthocyanin.
Response 6: Thank you for your comment. As suggested, we have revised title including TPC and TFC, this now appears in title of the revised manuscript. “Evaluation of anthocyanin profiling, total phenolic and flavonoid content, and antioxidant activity of Korean Rubus accessions for functional food applications and breeding”

Reviewer 3 Report
This research evaluated 15 Rubus accessions to identify superior resources with potential for use in functional foods and breeding. The topic is interesting. Generally, the manuscript is well organized and written well. The experimental design and results are reasonable. The references are appropriate.
The detailed comments are as follows:
- There is no need for two reference formats. For example, line 36 (Gevrenova et al., 2024) [1]. Delete one of them. Check the whole text.
- The novelty of this study should be clearly clarified in the last paragraph or the penultimate paragraph of Introduction.
- Make pictures in Table 1 clear. The rulers are blurred.
- Table 2 should be placed in Results.
- No statistical significance indicators for Table 4 and 5?
- Check reference styles. For example, no page numbers in Line 452.
Author Response
Reviewer 3
General comment: This research evaluated 15 Rubus accessions to identify superior resources with potential for use in functional foods and breeding. The topic is interesting. Generally, the manuscript is well organized and written well. The experimental design and results are reasonable. The references are appropriate.
Response: We sincerely thank the reviewer for the positive comments and thoughtful evaluation of our work. We are pleased to know that the overall organization of manuscript was well received. The detailed responses to specific suggestions and revisions are provided below.
The detailed comments are as follows:
Comment 1: There is no need for two reference formats. For example, line 36 (Gevrenova et al., 2024) [1]. Delete one of them. Check the whole text.
Check reference styles. For example, no page numbers in Line 452.
Response 1: Thank you for pointing this out. We have carefully reviewed the entire manuscript and removed duplicated reference format. All in-text citations and reference styles now appear in the appropriate numbered format only, as follow the journal guidelines.
Comment 2: The novelty of this study should be clearly clarified in the last paragraph or the penultimate paragraph of Introduction.
Response 2: Thank you for your valuable suggestion. We have revised the final paragraph of Introduction to more clearly emphasize the novelty of our study. This change is reflected in the revised manuscript, and this now appears as below.
At line 69 to 78: “Despite the growing interest in Rubus fruit research, comparative studies involving both wild and cultivated Rubus germplasms are still limited, particularly in East Asia. In addition, bioactive compounds responsible for antioxidant properties in Rubus fruits also closely related to therapeutic or pharmacological properties. These include an-ti-inflammatory, antimicrobial, anti-obesity, anticancer, and antidiabetic activities, which have been validated in both in vitro and in vivo models [5,10,11]. Such recent findings have elevated Rubus species as promising candidates not only functional food develop-ment but also for considerable pharmacological applications. Therefore, a deeper under-standing of their phytochemical and antioxidant properties is crucial for selecting elite accessions with superior biofunctional potential.”
At line 86 to 88: “To our knowledge, this is one of the few studies that systematically compares both wild and cultivated Rubus accessions from Korea.”
Comment 3: Make pictures in Table 1 clear. The rulers are blurred.
Response 3: Thank you for your valuable feedback. We have revised Table 1 to improve the clarity of the fruit images, adjusting the layout and size to enhance visibility. However, we acknowledge that the quality of the original photographs (low resolution via cellphone camera) was limited. Despite our efforts to enhance image sharpness, some limitations remain due to the initial image resolution at the time of sample collection. We will appreciate your understanding and consider acquiring high-resolution images in future studies.
Comment 4: Table 2 should be placed in Results.
Response 4: Thank you for your comment. As suggested, we have moved Table 2 to the Result section. This change is reflected in the revised manuscript, and the Table 2 now appears at the line 237, Page 6 of 17.
Comment 5: No statistical significance indicators for Table 4 and 5?
Response 5: Thank you for your comment. Table 4 presents the average IC50 values of each accession for four antioxidant assays, primarily for the purpose of ranking antioxidant performance. As this table focuses on comparative ranking rather than statistical groupings, we did not perform statistical significance tests on the IC50 values themselves. However, the individual antioxidant assay results with full statistical analyses are provided in Table S1 with legend in the supplementary materials. For this, we have added an expression “with average” in the revised manuscript at line 271.
In contrast, Table 5 includes statistical significance indicators and method in the table 5 legend (at line 278 to 279), which are now marked in the revised version. We hope this clarification addresses the concern.

Round 2
Reviewer 2 Report
The concerned issues have been addressed.
The concerned issues have been addressed.